# Sustainability of Traditional Rice Cultivation in Kerala, India—A Socio-Economic Analysis

**Jayasree Krishnankutty** [1], **Michael Blakeney** [2,*], **Rajesh K. Raju** [1] and **Kadambot H. M. Siddique** [3]

1.  Communication Centre, Kerala Agricultural University, Mannuthy, Kerala 680656, India; jayasree.krishna@kau.in (J.K.); rajkr369@gmail.com (R.K.R.)
2.  Faculty of Law, The University of Western Australia, Crawley, WA 6009, Australia
3.  The UWA Institute of Agriculture, The University of Western Australia, Perth, WA 6001, Australia; kadambot.siddique@uwa.edu.au
*   Correspondence: michael.blakeney@uwa.edu.au

**Abstract:** Traditional rice cultivars and cultivation are on the decline in most rice-growing areas, mainly as a result of their low productivity. Packed with nutritionally, environmentally and locally superior qualities, traditional cultivars hold the key for sustainability in rice cultivation. This study explored the dynamics of traditional rice cultivation in Kerala, India. It examined the economic, institutional and socio demographic factors involved in the production and marketing of traditional rice. We employed a multinomial logit model and discriminant function analysis to extract the key factors governing farmers' marketing behaviour, and various cost measures to study the economics of rice enterprises. The socio-demographic factors were analysed using descriptive statistical tools. Holding size and institutional support were the main factors governing the marketing behaviour of farmers. Even though traditional rice farming was not found to be cost-effective in implicit terms, it was remunerative when imputed personal labour and owned land costs were not considered. The study found that traditional farmers are ageing, have a lower education and use limited marketing channels. However, the majority of them were satisfied with their farm enterprise. By streamlining the market support mechanism and processing facilities, traditional rice would most likely gain momentum in key areas.

**Keywords:** traditional rice economics; institutional; socio-demographic factors; multinomial logit model; constraints

## 1. Introduction

In September 2000, world leaders at the United Nations Millennium Summit recognized a collective responsibility to work toward "a more peaceful, prosperous and just world" [1]. Following this Summit, the nations of the world committed to achieving certain goals across eight priority areas of social and economic development by 2015. Goal 1 of these millennium development goals (MDGs) was the eradication of extreme poverty and hunger. A stocktaking on the achievement of these goals undertaken by the United Nations Conference on Sustainable Development in Rio de Janeiro, June 2012, precipitated a process to develop a new set of Sustainable Development Goals (SDGs) intended to carry on the momentum generated by the MDGs beyond 2015 [2]. The SDGs sought to continue the fight against extreme poverty, but added the challenges of ensuring more equitable development and environmental sustainability. The UN General Assembly on 25 September 2015 adopted the 2030 Agenda for Sustainable Development, which in SDG2 aimed to "End hunger, achieve food security and improved nutrition and promote sustainable agriculture", and recognized the inter-linkages among supporting sustainable agriculture, empowering small farmers, promoting gender equality, ending rural poverty, ensuring healthy lifestyles and tackling climate change, among other issues [3].

Although promoted by the 2030 Agenda, the concept of sustainability is not a new concept and can be traced back at least to the 1987 report of the Brundtland Commission on Environment and Development, which defined sustainable development as that "which meets the needs of the present without compromising the ability of future generations to meet their own needs" [4]. This vagueness of meaning became transposed into the definition of sustainable agriculture. This has been described as having been formulated in a contested discourse [5] and the admitting of more than 70 definitions [6,7] suggest that it is an exercise in "measuring the immeasurable". Its multifarious meanings are described as open to value-judgment and leading to many different interpretations [8,9].

For our purposes we accept Professor Swaminathan's judgement that sustainable agriculture must achieve productivity in perpetuity without accompanying ecological and social harm [10] discussed in [11].

The forecast increase in the world population of up to 9.7 billion by 2050 and 10.9 billion in 2100 [12] will require a major effort to increase food production for the additional 2.6 billion people compared to today. Disregarding problems of distribution and food waste and loss, this would require an increase in global food production by at least one third, with the obvious potentially adverse ecological impacts which that might have.

A vigorous contemporary debate has been conducted around the sustainable agricultural strategies which might be adopted to meet the challenge of feeding the world's increasing population [13]. These range from precision farming [14] and the use of genetically modified crops [15–17] through to "ecogariculture" [18–21].

Declaring 2004 as the International Year of Rice, the UN General Assembly noted that rice is the staple food of more than half the world's population, and affirmed the need to heighten the awareness of the role of rice in alleviating poverty and malnutrition and reaffirmed the need to focus world attention on the role rice can play in providing food security and eradicating poverty (See http://www.fao.org/3/Y5167E/y5167e02.htm, accessed 7 December 2020.) During the Green Revolution of the 1960s, high yielding varieties (HYV) of rice were developed which increased yield, reduced the cropping period and increased cropping intensity to allow the cultivation of 2–3 crops per year, but which required the use chemical fertilizers, pesticides, tractors, mechanical threshers and controlled water supply to crops [22]. One of the major ecological consequences of the Green Revolution was the significant depletion in the number of traditional rice varieties, as the HYV had a very narrow and unstable genetic base compared with traditional varieties [23–26]. Traditional varieties have gradually disappeared as farmers have abandoned them in favour of monohybrid crops [27] In recent years, and in the face of climate change, it has been realised that traditional rice varieties represent a valuable gene pool for traits which may underpin the capacity of modern varieties of rice to adapt to climate change [28–30].

This article considers the contribution which the cultivation of traditional rice varieties can make to sustainable agriculture, by an examination of the socio-economic situation of farmers in traditional rice cultivation in Kerala.

Kerala lies in the south-western corner of the Indian peninsula, in the southern part of the Western Ghats adjoining Tamil Nadu and Karnataka in the east and north-east, and bounded by the Arabian Sea in the west. Kerala is topographically and ecologically diverse, consisting of a mix of coastland, wetlands and plains to the west, and the foothills of the Western Ghats to the east. The ecological conditions in the state have resulted in a considerable diversity of germplasm in both wild and cultivated rice [31]. It has a rich culture of rice cultivation, where rice farming is considered a symbol of prosperity and traditional lineage [32]. The rice requirements in this state are estimated as 3.5–4.0 million t/year. However, Kerala produces only one-fifth of this amount. The deficit in rice production is mounting each year, owing to the decline in the area under rice cultivation. Large-scale conversion of paddy lands for other crops or for residential purposes has caused a serious problem in the age-old practice of rice farming in Kerala [33].

The main reasons given for the decline in traditional rice varieties is their low productivity, longish growing duration, lack of price premium for some varieties and compara-

tively longer cooking time [34,35]. For many farmers, these factors outweigh the nutritive and environmental advantages of traditional rice varieties [36].

A more recent factor contributing to the decline in global rice production is climate change. Rainfed cultivation is estimated to account for about 25 per cent of global rice production, which makes it particularly vulnerable to fluctuations in rainfall, as well as heat stress from high temperatures [37,38]. With the expected demand for rice to increase in the coming years, food security will be imperilled, unless this situation can be improved. Farmers will have to increase yields by adopting high yielding varieties, or by utilising those traditional varieties which are suitable for marginal lands. A recent study [39] found that the cost of cultivating high yielding hybrid varieties in Assam was on average 29.43 percent higher than traditional rice. The higher costs were for plant protection chemicals (85.13 percent), irrigation (63.21 percent) and seed (62.81 percent). Under traditional rice cultivation, farmers used little or no plant protection chemicals, which makes cultivation environmentally sustainable, as well as economical [40].

## 2. Materials and Methods

Sustainability of an agricultural system is a time- and space-specific concept. Its assessment is closely linked to the context in which the specific farming system thrives [7]. Due to variations in biophysical and socioeconomic conditions, indicators used in one country may not be suitable for other countries and can be of great subjectivity [41]. The sustainability of traditional rice system will be analysed based on three broad dimensions in this study, which are adopted and adapted from the following:

Economic factors, especially cost of production and profitability—[42–47].
Socio demographic factors [48,49]—area; [50] experience; [15] age; [48,49]—education.
Institutional factors—[7,51].

Apart from production levels and profitability, of late, several authors [52,53] have postulated about their well-being as perceived by farmers, which is a derivative of their satisfaction in life, as a substantial indicator of sustainability. Therefore, we looked at the satisfaction level of traditional rice farmers, which will be a component to the system's sustainability.

With such a backdrop, the study addressed the following research questions in an attempt to link the answers to sustainability of traditional rice farming. (a) What is the varietal diverity of traditional rice in the study area? (b) How profitable is this rice system and how best the farmers market their produce? (c) What are the socio-demographic characteristics of the farmers cultivating traditional rice varieties and how do they tell on sustainability of the system? (d) How satisfied are the traditional rice farmers? (e) What are the constraints experienced by traditional rice farmers?

### 2.1. Study Area and Sampling

This study examined the cultivation of traditional rice varieties in the principal traditional rice-cultivating districts of Kerala. Three of the fourteen districts in Kerala state, namely Palakkad, Wayanad and Malappuram, had the largest area under traditional rice and were selected for the study. Palakkad is known as the rice bowl of Kerala, Wayanad is known for its hill area rice cultivation and Malappuram is rich with traditional, family rice farms. A random sample of 100 traditional rice farmers was chosen from each district, such that the total sample size for the study was 300. These farmers were cultivating traditional rice by choice, for consumption and marketing and were not government-supported farmers producing traditional varieties for conservation purposes.

### 2.2. Data Collection and Analysis

Farmer responses were collected through personal interviews using a semi-structured pre-tested interview schedule, focus group discussions and direct observation. The data generated were classified, coded, tabulated and analysed using a host of statistical tools. The socio-demographic characteristics of respondents were classified and interpreted using

simple descriptive statistical tools, including percentages and frequency distributions. Linear discriminant function analysis was undertaken as a multivariate test of differences between groups to determine the minimum number of dimensions needed to describe these differences. The influence of the explanatory variables on the marketing channel choices of farmers was explored in a multinomial logit model. The odds ratio was used to quantify the influence. A satiety index was developed to understand each farmer's level of satisfaction with traditional rice farming. Garrett's ranking analysis was used to prioritise farmer constraints.

## 2.3. Tools for Data Analysis

Cost Concepts Used in the Study

The cost concepts for farm management studies, used by the Commission on Agricultural Costs and Prices (CACP) of the Government of India, were employed. Data were collected on select physical indicators; the value of seed (purchased or home-grown), insecticides and fungicides, manure (owned and purchased), fertilisers, irrigation, machinery (own or hired machinery), human, animal and machine labour (own or hired), land revenue, rent paid for leased land or rental value of own land, interest on working capital, land revenue, depreciation of machinery and miscellaneous expenses.

The structure of different costs and their components [54] were as follows:

(i)  Cost $A_1$ includes value of human labour (casual and permanent), hired bullock power, owned bullock power, owned machine power, hired machine power, seeds (farm produced and purchased), manure (owned and purchased), fertilizer, plant protection chemicals, herbicides, irrigation charges, land tax (Landowners in India pay tax to the government; the levy is based on the area owned. Land tax is paid to the respective village office) and other taxes, depreciation on farm implements and buildings, interest on working capital and miscellaneous expenses;

(ii)  Cost $A_2$ = Cost $A_1$ + Rent paid for leased land;

(iii)  Cost $B_1$ = Cost $A_1$ + Interest on the value of owned fixed capital assets (excluding land);

(iv)  Cost $B_2$ = Cost $B_1$ + Rental value of owned land (less land revenue) and rent paid for leased land;

(v)  Cost $C_1$ = Cost $B_1$ + Imputed value of family labour;

(vi)  Cost $C_2$ (Cost of cultivation) = Cost $B_2$ + Imputed value of family labour;

(vii)  Cost $C_3$ = Cost $C_2$ + 10 percent of cost $C_2$ (to account for managerial input of the farmer).

## 2.4. Choice of Marketing Channels

A multinomial logit model (MNL) was used to quantify the predictors of variables that affect the marketing channels (0: consumption alone, 1: selling to friends and relatives; 2: selling to local markets; and 3: selling directly to Supplyco[2] at the farmgate) of traditional rice farmers of Palakkad, Malappuram and Wayanad. The MNL model assumes autonomy in the choice of conventional techniques for estimating multi-category dependent variables [55]. The chances of alternative marketing channel choices among farmers are shown below:

$$Prob = (Y_i = j) = \frac{\exp\left(\beta'_j x_i\right)}{\sum_{j=1}^{4} \exp\left(\beta'_j x_i\right)} \; for \; j = 0, 1, 2, 3 \qquad (1)$$

where:

$Y_i$ is the probability that farmers choose market $j$, $\text{pr}(Y_i = j)$;

$j = 0$: consumption alone; 1: selling to friends and relatives; 2: selling to local markets; 3: selling directly to Supplyco at the farmgate;

$x_i$ is the vector of households, production and marketing variables;

$\beta_j$ is the vector of coefficients associated with market choice $j$.

The odds ratio was estimated from the significant variables to identify the probability of improving the marketing channel choice

$$\text{Odds ratio} = \frac{Exp(B)}{1 + Exp(B)} \times 100 \qquad (2)$$

### 2.5. Farmer Satisfaction with Traditional Agriculture

Farmer satisfaction was measured by calculating the satiety index, which should act as an indicator of the sustainability of traditional rice (Supplyco is an institutional mechanism in Kerala operated by the government to procure produce from farmers at a predetermined price. Supplyco is an integral part of public distribution system in Kerala) cultivation. An arbitrary scale comprising questions on the probability of remaining in traditional farming, opinions about profitability, prices fetched by the produce, the procurement system was used, with the responses marked on a five-point continuum. The following formula was used for the satiety index:

$$Satiety\ index = \frac{S_i}{S_M} \times 100$$

where:

$S_i$ = Score obtained by the *ith* individual;
$S_M$ = Maximum possible score.

The satiety index categorised farmers as dissatisfied (<50), moderately satisfied (50–75) or highly satisfied (>75).

Furthermore, as with marketing channel choices, the MNL was used to measure the likelihood of improving farmer satisfaction with traditional agriculture.

### 2.6. Constraints in Traditional Rice Farming

Constraints faced by traditional rice farmers were analysed by using Garrett's ranking. Farmers were asked to express the constraints experienced in traditional farming via an open-ended question. The constraints expressed by the farmers were imputed into a frequency table. The rank of each constraint was converted to percent position by using the following equation:

$$\text{Per cent position} = \frac{100\left(R_{ij} - 0.5\right)}{N_j}$$

where:

$R_{ij}$ is the rank for *ith* constraint faced by the *jth* individual;
$N_j$ is the number of constraints ranked by the *jth* individual.

The rank obtained was an interval on a scale where its midpoint expressed the interval, such that 0.5 was subtracted from each rank. The percent position was converted into a score by using Garrett's table [56]. By using the score obtained from each constraint, the mean score was calculated and ranked according to the mean score.

## 3. Results and Discussion

### 3.1. Varieties under Cultivation

Despite having a government policy that does not support conversion of agricultural land for non-agricultural purposes, paddy lands are being used for other crops and non-agricultural purposes in Kerala state. As reported by the Kerala Bio Diversity Board, out of the nearly 160 rice varieties of Wayanad, 55 traditional varieties are now extinct [57,58]. A study conducted by Kerala Agricultural University and Dept of Agriculture and Farmers Welfare identified 63 cultivated traditional rice varieties in Wayanad district [59].

The major rice varieties grown in the study area are listed in Table 1. In Wayanad, the surveyed famers reported the highest productivity for variety "*Valichoori*" (>5000 kg/ha),

which is protected under the Protection of Plant Varieties and Farmers' Rights (PPVRFA). Regardless of their unique characteristics, the low productivity reported for varieties *Jeerakasala*, *Gandhakasala* and *Navara* were considered less preferred for widespread cultivation. In Palakkad, the main traditional cultivated varieties were *Chitteni*, *Chettadi*, *Thavalakkannan* and *Chenkazhama*, which form part of the registration of the geographical indication (GI) *Palakkadan Matta* under the Geographical Indications of Goods (Registration and Protection) Act 1999. These varieties were also grown in the Malppuram district. Most farmers in each region were unaware of their legal status with regard to cultivating these varieties.

**Table 1.** Registration status and area under cultivation of traditional varieties identified in the study area.

| Sl No | Variety | Registration Status | Area (ha) | Percentage of Total Area Surveyed | Year of Registration |
|-------|---------|---------------------|-----------|-----------------------------------|----------------------|
| 1 | Valichoori | Farmer Variety Reg No 221 | 24.93 | 20.87 | 2015 |
| 2 | Gandhakasala | Farmer Variety Reg No 57 Geographical Indication Certificate no. 34 | 2.15 | 1.80 | 2013 2010 |
| 3 | Jeerakasala | Farmer Variety Reg No 59 Geographical Indication Certificate no. 34 | 1.2 | 1.00 | 2013 2010 |
| 4 | Adukkan | Farmer Variety Reg No 23 | 12.15 | 10.17 | 2016 |
| 5 | Navara | Geographical Indication Certificate no. 17 | 3.31 | 2.77 | 2007 |
| 6 | Mullankaima | Farmer Variety Reg No 220 | 0.20 | 0.17 | 2015 |
| 7 | Chomala | Farmer Variety Reg No 59 | 0.20 | 0.17 | 2013 |
| 8 | Chitteni * | Geographical Indication Certificate No 40 | 33.99 | 28.45 | 2013 |
| 9 | Rakthasali | | 2.90 | 2.43 | |
| 10 | Thavalakkannan * | Geographical Indicator Certificate No 40 | 2.67 | 2.24 | 2013 |
| 11 | Thondi | Farmer Variety Reg No. 61 | 12.53 | 10.49 | 2013 |
| 12 | Chettadi * | Geographical Indicator Certificate No 40 | 20.5 | 17.16 | 2013 |
| 13 | Chenkazhama * | Geographical Indicator Certificate No 40 | 1.21 | 1.01 | 2013 |
| 14 | Kattamodan | | 0.70 | 0.59 | |
| 15 | Kochumannan | | 0.21 | 0.18 | |
| 16 | Vella kayama | | 0.21 | 0.18 | |
| 17 | Thekkancheera | | 0.40 | 0.33 | |
| | Total | | 97.73 | 100 | |

Note: * The varieties under GI Certificate 40 are together labelled as Palakkadan matta and not as their individual names.

Traditional rice cultivation is declining in Kerala, as evidenced by the total area sown by the 300 surveyed farmers (97.73 ha). Legal and governmental measures exist to protect and support the traditional systems, but neither appear to be making a difference for farmers. In most cases, the farmers are unaware of the legal support mechanisms in place for their cultivated variety. For example, one reason for the low marketing efficiency of traditional rice is that farmers sell it as raw rather than de-husked grains because of the lack of suitable milling facilities for traditional rice.

The 300 surveyed respondents cultivated a total area of 97.93 ha of traditional rice (average 0.33 ha per farm), which is a true reflection of the very low area under traditional rice cultivation. This can be indicative of the substantial decline in varietal diversity in practice, because, by default, *padasekharams* (a collection of rice fields owned by different farmers) in Kerala have at most two or three varieties. Farmers keep this uniformity for ease and efficiency in management. Our survey results showed only 17 varieties being cultivated for consumption or marketing in the study area. This indicates a clear and ominous erosion of the gene pool of valuable traits.

### 3.2. Respondent Profile–Socio Demographic Characteristics

Table 2 shows that most of the surveyed respondents were 40–70-years-old (81%), male (87%), and had, at most, a high school education (77%), more than 30 years farming

experience (63%) and cultivated less than 1 ha of land. Aging of traditional farmers has been reported in other rice-growing parts of the world. [60].

**Table 2.** Distribution of respondents according to socio-demographic characteristics.

|  |  | Palakkad (n = 100) | Malappuram (n = 100) | Wayanad (n = 100) | Total (N = 300) |
|---|---|---|---|---|---|
|  |  | **Age** |  |  |  |
|  | <40 | 2 | 13 | 16 | 31 (10.33) |
|  | 40–54 | 34 | 33 | 38 | 105 (35.0) |
| **Frequency** | 54–69 | 49 | 49 | 38 | 136 (45.33) |
|  | >70 | 15 | 5 | 8 | 28 (9.33) |
|  | Total | 100 | 100 | 100 | 300 (100) |
|  |  | **Gender** |  |  |  |
|  | Male | 84 | 92 | 85 | 261 (87.00) |
| **Category** | Female | 16 | 8 | 15 | 39 (13.00) |
|  | Total | 100 | 100 | 100 | 300 (100) |
|  |  | **Education** |  |  |  |
|  | Primary | 41 | 36 | 58 | 135 (45.00) |
|  | High school | 32 | 38 | 28 | 98 (32.67) |
| **Category** | SSLC and above | 20 | 18 | 8 | 46 (15.33) |
|  | College and above | 7 | 8 | 6 | 21 (7.00) |
|  | Total | 100 | 100 | 100 | 300 (100) |
|  |  | **Experience** |  |  |  |
|  | <15 | 5 | 14 | 4 | 23 (7.67) |
|  | 15—30 | 37 | 23 | 28 | 88 (29.33) |
| **Frequency** | 30–45 | 36 | 44 | 38 | 118 (39.33) |
|  | >45 | 22 | 19 | 30 | 71 (23.67) |
|  | Total | 100 | 100 | 100 | 300 (100) |
|  |  | **Area(ha)** |  |  |  |
|  | <0.404 | 57 | 59 | 41 | 157 (52.33) |
|  | 0.404–0.809 | 31 | 33 | 39 | 103 (34.33) |
| **Category** | 0.809–1.21 | 10 | 3 | 12 | 25 (8.33) |
|  | >1.21 | 2 | 5 | 8 | 15 (5.00) |
|  | Total | 100 | 100 | 100 | 300 (100) |

This picture of an ageing, less educated, experienced body of small holder farmers is similar in most of parts of the rice cultivating world [61] and these are the farmers who keep up traditional rice systems. This poses a serious question in terms of the sustainability of these farming systems in eras to come.

*3.3. Economics of Traditional Rice Farming*

Economic viability is a major consideration for sustainable farming. The total rice area under cultivation in 2017–18 was 75,415 ha, 7864 ha and 8026 ha in Palakkad, Malappuram and Wayanad districts, respectively, or 47 per cent of the total rice paddy area in Kerala [62]. These three districts contribute 46.8 per cent of the total rice production in Kerala, with production higher than the state average in Malappuram, lower than the state average in Palakkad and Wayanad. The costs of cultivation and production in 2016–2017 was Rs. 112,862/ha and Rs.31.40/kg, respectively (Table 3; [63].

**Table 3.** Area, production, and costs of cultivation and production of rice in Kerala.

| Districts | Area (ha) (2017–18) | Production (tonnes) (2017–18) | Productivity (kg/ha) (2017–18) | Cost of Cultivation C) (Rs/ha) (2016–17) | Cost of Production (Rs/kg) (2016–17) |
|---|---|---|---|---|---|
| Palakkad | 75,415 | 198,626 | 2633 |  |  |
| Malappuram | 7864 | 23,571 | 2999 |  |  |
| Wayanad | 8026 | 21,792 | 2715 |  |  |
| Kerala | 194,235 | 521,310 | 2757 | 112,862 | 31.40 |

Source: Agricultural statistics 2017–18 Department of Economics and Statistics, GoK. Report on cost of cultivation of important crops in Kerala 2016–17, Department of Economics and Statistics, GoK.

The calculation for the cost of farming of traditional rice in Palakkad, Malappuram and Wayanad was based on three cost concepts detailed in Methods. Cost A in general

covers the total paid up costs by the farmer, Cost B the interest and rental values, and Cost C the imputed values and imputed managerial costs. The calculation outcomes for each district are in Table 4. Since no land was rented in the study area, Cost $A_1$ and $A_2$ were the same; thus, Cost $A_1$ and $B_1$ were the same as the only fixed resources was land. The workers used their own equipment, and were paid accordingly in their wages. The Malappuram district had the highest values for Cost $C_1$, $C_2$, and $C_3$ and the Wayanad district had the lowest.

**Table 4.** Cost of cultivation and cost of production of traditional rice in the study area.

| | Cost of Cultivation (Rs./ha) | | | | Cost of Production (Rs./kg) | | |
|---|---|---|---|---|---|---|---|
| Districts/Cost | Palakkad | Malappuram | Wayanad | Districts/Cost | Palakkad | Malappuram | Wayanad |
| Cost $A_1$ | 50,806 | 52,181 | 51,063 | Cost $A_1$ | 18.51 | 19.31 | 15.39 |
| Cost $A_2$ | 50,806 | 52,181 | 51,063 | Cost $A_2$ | 18.51 | 19.31 | 15.39 |
| Cost $B_1$ | 50,806 | 52,181 | 51,063 | Cost $B_1$ | 18.51 | 19.31 | 15.39 |
| Cost $B_2$ | 68,306 | 74,181 | 68,563 | Cost $B_2$ | 25.14 | 27.43 | 20.71 |
| Cost $C_1$ | 57,374 | 56,525 | 54,622 | Cost $C_1$ | 22.94 | 21.57 | 17.06 |
| Cost $C_2$ | 74,874 | 78,525 | 72,122 | Cost $C_2$ | 29.57 | 29.69 | 22.38 |
| Cost $C_3$ | 82,361 | 86,378 | 79,334 | Cost $C_3$ | 32.53 | 32.66 | 24.62 |
| | Average yield/ha | | | | 2675 | 2877 | 3970 |

The cost of cultivation in all three districts was much lower than the state average for small-holding rice (Rs 112,862/ha, DoES, 2017) making traditional rice cultivation an attractive option in these areas. The rental value of owned land escalated Cost $C_3$, resulting in a slightly higher than average value (Rs. 31.40) except for the Wayanad district.

The Malappuram and Wayanad districts had higher productivity than the state average (2757 kg/ha), Wayanad had the highest productivity and Palakkad had the lowest (Table 4). The higher productivity in Wayanad can be attributed to the variety *Valichoori* which yields >5000 kg//ha, as reported by the farmers.

Different measures of income were used to identify the economic viability of traditional rice farming in the Palakkad, Malappuram and Wayanad districts (Table 5). According to the survey response, farmers in Wayanad had the highest average gross income (Rs. 85,281/ha), while those in Palakkad had the lowest (Rs. 71,578/ha). Similarly, Wayanad had the highest farm business income (Rs. 34,218/ha) and Palakkad had the lowest (Rs. 20,772/ha). Three speciality rices—*Navara* (medicinal variety) and *Gandhakasala* and *Jeerakasala* (aromatic varieties)—grow best in Wayanad, generating a higher price than other varieties, which would explain the higher farm business income in Wayanad.

**Table 5.** Estimates of different measures of income (Rs./ha).

| Measures | Palakkad | Malappuram | Wayanad |
|---|---|---|---|
| Gross income (GI) | 71,578 | 7936 | 85,281 |
| Farm business income (GI-Cost $A_1$) | 20,772 | 27,147 | 34,218 |
| Family labour income (GI-Cost $B_2$) | 3272 | 5144 | 16,718 |
| Net Income (GI-Cost $C_3$) | −10,783 | −7054 | 5947 |
| BC (GI:$C_3$) | 0.87 | 0.91 | 1.07 |
| BC At Explicit (GI:$A_1$) | 1.40 | 1.52 | 1.67 |

The net income and benefit:cost ratio indicated that farming was a loss-making business for respondents in Palakkad and Malappuram if family labour and land values were considered. In contrast, the positive net income and benefit:cost ratio in Wayanad was attributed to the high productivity of *Valichoori* and the higher price fetched by varieties

*Jeerakasala* and *Gandhakasala*. The benefit:cost ratio for explicit cost (i.e., Cost $A_1$) was >1 for all districts, with the highest in Wayanad (1.67) followed by Malappuram (1.52) and Palakkad (1.40).

This led us to consider the marketing behaviour of traditional rice farmers, with many rectifiable gaps identified.

### 3.4. Marketing Channels

Marketing is a concern for farmers, irrespective of the crops they cultivate. A common feature of grain marketing systems in developing countries is the co-existence of a government marketing agency (parastatal) and a parallel private marketing channel with many private intermediaries. These parastatals are assigned to control or regulate the system [64]. The survey revealed that most farmers sell their produce through the state-owned procurement and distribution agency, the parastatal in Kerala, named Supplyco, for a pre-determined price (Rs.25.3/kg at the time of the study).

In Wayand three marketing channels were identified (Figure 1A), of which Supplyco dominated with 52.46 per cent of the marketed volume. However, Supplyco, had limited penetration in some regions, such that farmers were forced to sell through other channels for a lower price (Table 6). Only varieties such as *Jeerakasala* and *Gandhakasala* obtained a higher price owing to their unique characteristics.

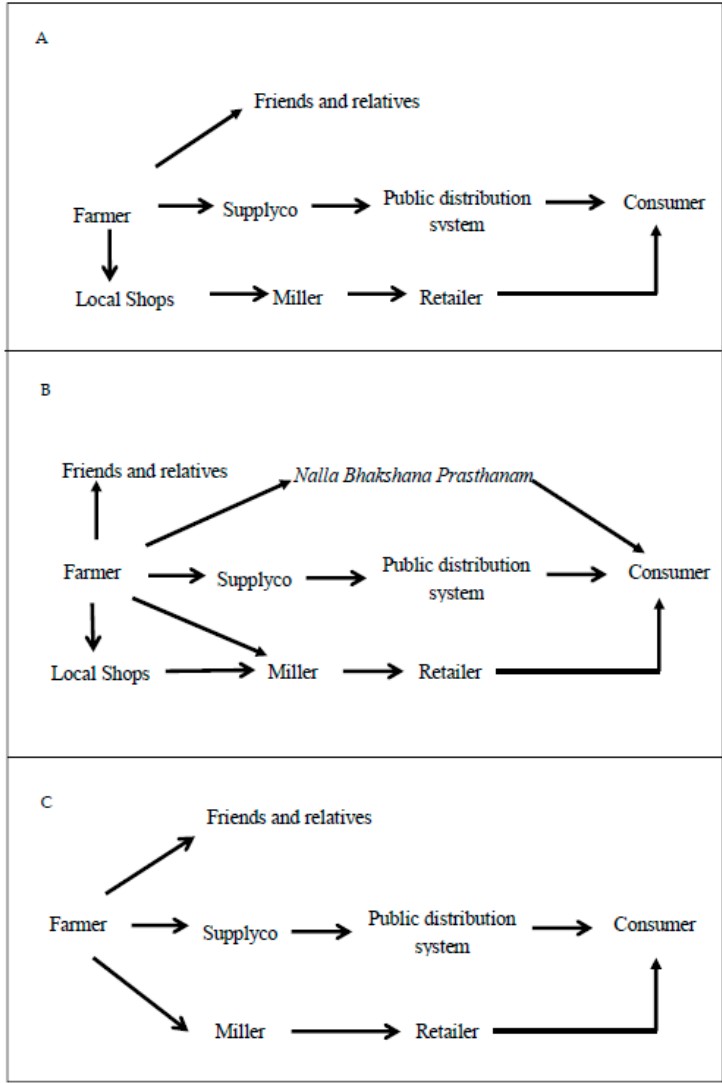

**Figure 1.** Marketing channels in Wayanad, Malappuram and Palakkad.

**Table 6.** The volume of rice marketed through each channel in the three districts.

| Districts | Marketing Channels | Volume (%) |
|---|---|---|
| Wayanad | Channel 1 | 37.6 |
| | Channel 2 | 52.46 |
| | Channel 3 | 9.94 |
| | TOTAL | 100 |
| Malappuram | Channel 1 | 4.04 |
| | Channel 2 | 22.73 |
| | Channel 3 | 59.44 |
| | Channel 4 | 4.52 |
| | Channel 5 | 9.27 |
| | TOTAL | 100 |
| Palakkad | Channel 1 | 8.68 |
| | Channel 2 | 79.37 |
| | Channel 3 | 11.95 |
| | TOTAL | 100 |

In Malappuram, five marketing channels were identified (Figure 1B), with Supplyco having the highest share of marketed volume (Table 6). *Nalla Bhakshana Prasthanam*, an NGO work to provide "*safe to eat*" products, and is another rice marketing channel. Members of the society chose this avenue, as they received a higher price than at Supplyco. Farmers cultivating varieties such as *Navara* and *Rakthasali* were doing contract farming with a local miller who provided the seeds and procured the produce at a higher price than the prevailing local market price.

In Palakkad, Supplyco had the greatest marketing share of the marketed volume (almost 80%). The other avenues were selling to millers and friends and relatives (Figure 1C, Table 6).

The greatest quantity of rice is transacted through Supplyco; however, its performance had been criticised [65] since its establishment. Procurement lags and delays in payment often put farmers under stress [66]. The farmers only receive a standard price for traditional rice, as fixed by the agency, irrespective of the variety.

To fit the multinomial logit model, the farmers were classified into three categories according to their marketing behavior, such as, using their produce for consumption alone, selling in local markets and selling through Supplyco. These were considered as the first, second and third stages (Table 7). The significant variables at each stage indicate that they are the explanatory variables for farmers' progress from one stage to the other. In the first stage, age, education, area cultivated and yield were the decisive variables for their transition to the next. Interestingly, the awareness of marketing options can essentially promote their move from local markets to Supplyco, that is, from the second to the third stage. Therefore, enhancing the institutional support for marketing itself can help farmers improve their marketing. Education and production amount were identified as the main factors governing farmers' choice of marketing channels [67] attitude to risk, asset ownership, institutional variables, transaction costs and market attributes [68] and age of household, education status, credit access, off-farm income and total land-holding size [69].

To assess model fit in MLR, the most commonly used tool is the likelihood ratio test. Significance at less than 0.05 suggests a model fit [70]. To evaluate the goodness-of-fit of logistic models, several pseudo R-squared models have been developed. One method that is endorsed repeatedly [71,72] is the one proposed by [73]. According to McFadden, pseudo $R^2$ values from 0.2–0.4 indicate excellent model fit.

Marketing options will improve as institutional support improves and yields increase as well as the area cultivated by farmers (Table 8). In other words, these three factors differentiate the marketing behavior of farmers in the three districts, which were also significant in the case of potato farmers [74].

**Table 7.** Variables explaining farmers' choice of marketing channels—multinomial logit model.

| Marketing Channels | Parameters | Odds Ratio | Chance of Improvement |
|---|---|---|---|
| 0–1 | Age | 0.580 | 36.69 |
| | Education | 0.678 | 40.42 |
| | Area | 0.037 | 3.53 |
| | Yield | 0.999 | 49.99 |
| 1–2 | Area | 0.318 | 24.15 |
| | Yield | 0.999 | 49.98 |
| 2–3 | Age | 0.587 | 36.98 |
| | Education | 0.663 | 39.88 |
| | Mobility | 0.621 | 38.29 |
| | Yield | 0.999 | 49.98 |
| | Awareness | 2.263 | 69.36 |

| **Model Fitting Information** | | | | |
|---|---|---|---|---|
| Model | Model Fitting Criteria | Likelihood Ratio Tests | | |
| | −2 Log Likelihood | Chi-Square | df | Sig. |
| Intercept Only | 708.760 | | | |
| Final | 546.513 | 162.247 | 27 | 0.000 |

| **Pseudo R-Square** | |
|---|---|
| McFadden | 0.228 |

**Table 8.** Standardized Canonical Discriminant Function Coefficients for the choice of marketing channel.

| Variable | Function |
|---|---|
| | 1 |
| Institutional support | 0.897615 |
| Yield | 0.359001 |
| Area | 0.348232 |

### 3.5. Are Farmers Satisfied?

There are concerns about the economic viability and sustainability of traditional rice varieties. One study reported that the yields of traditional and modern rice varieties do not differ significantly [75], while another concluded that farmers rely on modern varieties because of their higher yields [76]. Traditional landraces are often more resilient to trying environmental conditions and produce more reliable yields across many situations than modern varieties [76]. We evaluated the farmers' satisfaction with probability of continuing with traditional rice and this was calculated as a 'satiety index' (Table 9).

**Table 9.** Percentage distribution of farmers according to satiety index.

| Category | Per Cent (N = 300) |
|---|---|
| >50 | 10 |
| 50–75 | 66.67 |
| 75–100 | 26.33 |
| **Grand total** | **100** |

Of the 300 survey respondents, only 10 per cent were dissatisfied with traditional rice farming, while 26.33 were highly satisfied, which is a promising result for the sustainability of traditional rice cultivation. Furthermore, we evaluated the possibility of improving the satisfaction level using a multinomial logit model and calculated the odds ratio (Table 10).

The multinomial logit model showed that age, education, years of experience, mobility (access to the market), cultivated area and yield are significantly linked with farmer satisfaction in the first tier. Improvement in any of these variables would improve the satisfaction level of farmers in category I (dissatisfied) to category II (moderately satisfied).

Education, cultivated area [77], years of experience and mobility (access to transport) were the factors responsible for maximizing the farmers' satisfaction, i.e., in the second tier.

**Table 10.** Odds ratio of the satiety index of farmers in traditional rice cultivation with tables.

| Satiety Index | Parameters | Odds Ratio | Chance of Improvement (%) |
|---|---|---|---|
| 1–2 | Age | 0.418 | 29.49 |
| | Education | 0.591 | 37.16 |
| | Years of experience | 1.793 | 64.20 |
| | Mobility | 0.709 | 41.49 |
| | Area | 1.423 | 58.73 |
| | Yield | 1.000 | 50.01 |
| 2–3 | Age | 0.476 | 32.25 |
| | Education | 0.812 | 44.81 |
| | Year of experience | 1.469 | 59.49 |
| | Mobility | 0.649 | 39.35 |
| | Area | 1.420 | 58.67 |
| **Model Fitting Information** | | | |
| Model | Model Fitting Criteria | Likelihood Ratio Tests | |
| | −2 Log Likelihood | Chi-Square | df | Sig. |
| Intercept Only | 708.760 | | | |
| Final | 545.697 | 163.063 | 30 | 0.000 |
| **Pseudo R-Square** | | | |
| McFadden | | 0.229 | |

*3.6. Farmers' Constraints*

Table 11 summerises the constraints expressed by the traditional rice farmers. Most of them (82%) do not rely on milling because they sell the rice as raw grains. Factors including grain shape (length:width ratio) and hardness affect milling efficiency. Round grains (raw) with low length:width ratios are difficult to break, while slender grains with higher ratios are easy to break. The surface hardness of the brown rice kernel is a varietal characteristic that determines the extent to which the grain can resist the forces applied during milling. Lower surface hardness facilitates breakage during milling, thus reducing milled rice recovery and quality [78]. High hulling percentages increase the recovery of rice [79]. Traditional rice varieties are more prone to breakage during milling and low hulling percentages than modern varieties, which reduces their commercial value and deters wholesalers from procuring traditional rice. Farmers who try to market traditional rice alone are aware of this drawback. Subsequently, traditional rice sold to wholesalers is mixed with other (modern) rice and sold under brand names that do not identify the rice varieties. Most urban consumers buy this type of rice.

**Table 11.** Constraints faced by the respondents.

| Sl No. | Constraints | Mean Score | Rank |
|---|---|---|---|
| 1 | Shortage of skilled labour | 64.64 | 1 |
| 2 | Delay in payment | 54.72 | 2 |
| 3 | Shortage of water/rain | 53.48 | 3 |
| 4 | Lack of institutional support | 53.09 | 4 |
| 5 | Low productivity of labour | 48.89 | 5 |
| 6 | Transportation facility | 47.53 | 6 |
| 7 | Neighbourhood practices | 46.09 | 7 |
| 8 | Lack of milling facility | 43.93 | 8 |
| 9 | Animal attack | 39.87 | 9 |

## 4. Conclusions

The drivers of agricultural sustainability in developing countries has been observed to encompass a range of demographic, natural, socio-economic, political, institutional and management factors [80]. Traditional rice cultivation could be a case study for sustainable agriculture. It has lower costs of cultivation than modern varieties because traditional varieties have evolved locally and have thrived for generations, resulting in fewer pest and disease issues and the ability to withstand climatic variations. Their medicinal, nutritive and safety values are considerable [23]. This study shows that in explicit terms traditional rice cultivation is less costly, which is a point of further thought for their promotion on larger scale in developing countries of a comparable situation.

The study supports the observation of [30] that farmers' decisions are influenced to a large extent by socioeconomic factors and that holding size, education status and yield influenced cultivation decisions. Because of the value of traditional varieties to sustainable agriculture and on-farm conservation, innovative government support policies are counselled to strengthen and sustain the traditional rice system. However, as [80] have observed, government policies to promote the cultivation of traditional rice varieties in the Western Ghats, by the promotion of on-farm conservation of crop genetic resources has been ineffective, because of an absence of financial incentives and education as to how these varieties satisfy their livelihood concerns, such as avoidance of risk, yield maximization, input suitability, yield stability and tolerance to environmental stress and marketability. The current study echoes those results in counselling educational extension activities by the agricultural authorities [81]. It highlights the need for consistent and timely institutional support in marketing.

Holding size and institutional support were the main factors governing the marketing behaviour of farmers. These promoted them to look for profitable marketing avenues. Even though traditional rice farming was not found to be cost-effective in implicit terms, it was remunerative when imputed personal labour and owned land costs were not considered. For the farmers involved in traditional rice cultivation, the strict economic validations of these factors did not matter so much, as they held their farming as more of a cultural heritage. The study also found that traditional farmers are ageing, have lower education and use limited marketing channels. But still, the majority of them were satisfied with their farm enterprise, which poses a positive note on sustainability. The reason for this might be that most traditional rice growers were also traditional in nature. The concept of the relative advantage in growing a potential high value crop seems to be lost on them and this is one area where concentrated awareness generation is necessary, as many of the respondents of the study said that they raised the traditional varieties only as a continuation of ancestral practice. This reality prevails in other Asian rice cultivating countries which see an erosion of young generation from rice farming, especially traditional rice farming. Thus, this study calls for a continuing dialog between the extension, research and policy systems, which should lead to the next generations being educated and supported in holding up the intrinsic value of these heritage crops and their sustainability at a time of climate change.

**Author Contributions:** J.K. wrote the report of the study. M.B. provided the mateial on the agricultural policy context. R.K.R. co-ordinated the surveys of farmers and relevant stakeholders. K.H.M.S. provided an over-view of all contributions. All authors have read and agreed to the published version of the manuscript.

**Funding:** This research was funded by the Australian Research Council, grant number DP170100747.

**Institutional Review Board Statement:** The study was conducted according to the guidelines of the Declaration of Helsinki, and approved by the Ethics Committee of the University of Newcastle (protocol code H-2017-0054 date of approval 1 May 2019).

**Informed Consent Statement:** Informed consent was obtained from all subjects involved in the study.

**Data Availability Statement:** The data presented in this study are available on request from the corresponding author.

**Conflicts of Interest:** The authors declare no conflict of interest.

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
