# Peer review of "Sustainability of Traditional Rice Cultivation in Kerala, India—A Socio-Economic Analysis"

_sustainability, doi:10.3390/su13020980_

Round 1
Reviewer 1 Report
The results need to be revised and modified.
The authors include the following aspects in the results (and discussion):
- registration status and area under cultivation of traditional varieties identified in the study area
- distribution of respondents according to socio demographic characteristics
- area, production, and costs of cultivation and production of rice in Kerala
- cost of cultivation and cost of production of traditional rice in the study area.
- estimates of different measures of income
- the volume of rice marketed through each channel in the three districts
- variables explaining farmers’ choice of marketing channels – Multinomial logit model
- standardized Canonical Discriminant Function Coefficients for the choice of marketing channel
- percentage distribution of farmers according to satiety index
- odds ratio of the satiety index of farmers in traditional rice cultivation with iables
- constraints faced by the respondents
It is not clear and evident, what are the characteristics of the study area and sample, and what are the main results of the analysis conducted that add novelty and relevant discussions for the case study and for the literature.
Author Response
Thanks for your suggestion. Please see the attachment.

Reviewer 2 Report
The paper has undergone substantial improvements. Therefore, I think that it is now suitable for publication in Sustainability journal. I would like the authors to respond to the following comments before submitting the final manuscript.
1) Please close the Introduction section with your research questions, as now it provides very little information about the scope of the paper. You can move the research questions from the Materials and methods section to the Introduction.
2) Please, provide the results of any diagnostic test for the goodness of fit of the Multinomial logistic models. I would suggest indicating at least the results of the Likelihood ratio test and the Pseudo R2 values.
Author Response

(The authors gave the same response as above.)

Round 2
Reviewer 1 Report
As already commented, as far as I am concerned it is not clear why the paper contains all the following results (and discussion):
- registration status and area under cultivation of traditional varieties identified in the study area
- distribution of respondents according to socio demographic characteristics
- area, production, and costs of cultivation and production of rice in Kerala
- cost of cultivation and cost of production of traditional rice in the study area.
- estimates of different measures of income
- the volume of rice marketed through each channel in the three districts
- variables explaining farmers’ choice of marketing channels – Multinomial logit model
- standardized Canonical Discriminant Function Coefficients for the choice of marketing channel
- percentage distribution of farmers according to satiety index
- odds ratio of the satiety index of farmers in traditional rice cultivation with iables
- constraints faced by the respondents
In other words, all the 11 tables refer to relevant results from the study conducted? Some seem to be description data
Round 3
Reviewer 1 Report
none
This manuscript is a resubmission of an earlier submission. The following is a list of the peer review reports and author responses from that submission.
Round 1
Reviewer 1 Report
Thank you for providing me with the opportunity to review the paper “Sustainability of traditional rice cultivation in Kerala, 2 India–A socio economic analysis”. The authors have done a great quantitative analysis but the paper lacks the proper theoretical background to be considered as a research paper with implications for readers without any knowledge or interest for the considered area. Therefore, I have to propose its rejection. I would be very happy to review again the paper, provided that the authors consider the following comments.
1) The introduction lacks any theoretical discussion that would emphasize the added value of the paper. Therefore, the authors should elaborate more on issues of how traditional rice cultivation is related to sustainability. A literature review on this relationship should be firstly conducted, considering aspects, such as the need to protect traditional cultivations, locals’ way of life, sustainable traditional value chains and so forth. A connection with wider sustainability targets, such as the UN sustainable development goals could be very helpful in making the paper more solid in theoretical terms. This would highlight the need for conducting their analysis.
2) The empirical part should be founded on a theoretical/conceptual model of measuring sustainability. Authors should provide more evidence on how their empirical analysis is related to sustainability. Therefore, they should first show how the issues addressed by their analysis, such as cost, life satisfaction, marketing, etc, are linked with the sustainability of the sector and the local society in general. Therefore, a literature review on how other researchers dealt with the sustainability of the considered sector or of other traditional cultivations would be essential. By doing that, the authors will be able to demonstrate how their approach differs from the existing ones and how they extend our knowledge not only for the Kerala case but also for the global target of achieving sustainability.
3) The conclusions section should discuss the wider implications of the paper and not only those referring to the Kerala case. I believe that if they incorporate the preceding comments into the revised manuscript, then this would be a rather easy task.
Reviewer 2 Report
The research conducted seems valid and interesting. However, the paper is written in an unclear way, badly organized and the objectives (related to methodology and experimentation) are not explicit.
Following specific comments for the various parts of the paper.
The objectives should be highlighted in more detail in the Introduction, placing them in the context of other research conducted, highlighting the novelties of the research.
The Methodology section must be rewritten, creating a "Material and Methods" paragraph, inserting the data used and the methodology. I can't understand why the data on study area, respondents, production and costs have been included in the results part.
The Results section could consider only the estimation results on the marketing channels and then discuss these results in more depth, making a comparison with other studies, discussing economic and political implications and so on.
The Conclusions repeat concepts already said previously and must be rewritten in greater depth, highlighting strengths, weaknesses, further developments.